# Is body fat mass associated with worse gross motor skills in preschoolers? An exploratory study

**Juliana Nogueira Pontes Nobre**[1]*, **Rosane Luzia De Souza Morais**[1], **Amanda Cristina Fernandes**[1], **Ângela Alves Viegas**[1], **Pedro Henrique Scheidt Figueiredo**[2], **Henrique Silveira Costa**[2], **Ana Cristina Resende Camargos**[3], **Marco Fabrício Dias-Peixoto**[1], **Vanessa Amaral Mendonça**[2], **Ana Cristina Rodrigues Lacerda**[2]

1 Centro Integrado de Pós-Graduação e Pesquisa em Saúde (CIPq-Saúde), Universidade Federal dos Vales do Jequitinhonha e Mucuri (UFVJM), Diamantina, Minas Gerais, Brazil, 2 Faculdade de Fisioterapia, Universidade Federal dos Vales do Jequitinhonha e Mucuri (UFVJM), Diamantina, Minas Gerais, Brazil, 3 Faculdade de Fisioterapia, Universidade Federal de Minas Gerais (UFMG), Belo Horizonte, Brazil

* junobre2007@yahoo.com.br, junpnobre@gmail.com

## Abstract

We compared the motor competence between overweight/obese and eutrophic preschoolers with similar physical activity levels, age, socioeconomic status, maternal education, quality of the home environment and quality of the school environment. We also investigated to what extent excess body fat mass explains gross motor skills in preschoolers. A cross-sectional quantitative and exploratory study was conducted with 48 preschoolers assigned into eutrophic and overweight/obese groups. Overweight/obese preschoolers had worse Locomotor subtest standard scores than the eutrophic ones ($p = 0.01$), but similar Object Control subtest and Gross Motor Quotient scores ($p > 0.05$). Excess body fat mass explained 12% of the low Locomotor subtest standard scores in preschoolers ($R^2 = 0.12$; $p = 0.007$). Excess body fat mass was associated with worse locomotor skills when the model was adjusted for physical activity levels, age, socioeconomic status, maternal education, quality of the home environment and quality of the school environment. Thus, excess body fat mass partly explains lower locomotor skills in preschoolers.

## Introduction

Childhood obesity is a growing public health concern worldwide [1]. In the past decades, childhood obesity was considered a problem, especially in developed countries, but recent evidence indicates the rapid growth of obesity in children from developing countries. [2, 3]. Epidemiological data point to the comorbidities related to excess body fat mass, such as impaired glucose tolerance and hepatic steatosis [4], as the main responsible for the childhood obesity pandemic [1, 5]. In Brazil, among children under 5 years of age, 14.8% are overweight and 7% are obese [6].

**Funding:** This study was financed (financial support and scholarships) in part by the Fundação de Amparo à Pesquisa de Minas Gerais (FAPEMIG APQ-01898-18), Conselho Nacional de Desenvolvimento Científico e Tecnológico (CNPq), the Coordenação de Aperfeiçoamento de Pessoal de Nível Superior - Brasil (CAPES) - Finance Code 001, and the Universidade Federal dos Vales do Jequitinhonha e Mucuri (UFVJM). The funders had no role in study design, data collection and analysis, decision to publish, or preparation of the manuscript.

**Competing interests:** The authors have declared that no competing interests exist.

Early childhood, especially the preschool period between 3 to 5 years [7], is the stage of life when the body mass index (BMI) is reduced to a minimum physiological value known as adiposity rebound [8]. Studies indicate that weight gain during preschool years (but not during school years) is associated with a risk of overweight or obesity in adolescence, a factor that characterizes this phase as sensitive for preventing excess body fat mass [8]. Moreover, it is also considered a sensitive phase for learning fundamental movements; it is a time to expand the motor repertoire and experience the movements that will contribute to the development of future skills that will evolve into sports or leisure practices [9]. Studies indicate the preschool period as the critical moment for the development of motor skills and healthy behaviors that may influence physical activity (PA) practices throughout life [10, 11].

Regarding the impact of childhood obesity on the development of children, a previous study of our team, including 6 to 24 months overweight and obese infants, evidenced that although overweight and obese infants presented a development within a normal range, they had a worse motor and cognitive skills [12]. Of note, the analyses performed in this previous study considered factors that could interfere with child development, e.g., socioeconomic status, mother education, resources of the domestic and yard environment [12].

Although previous work has shown that fat mass was not related to any motor skill measure [13], other studies have found an association between overweight/obesity with low motor competence (MC) in children [10, 12, 14–16], including a work with Brazilian preschoolers from the Northeast region of Brazil [15] and longitudinal studies [17–19]. However, to our knowledge, there remains a gap regarding the association between overweight and obesity with motor competency, specifically in children with preschool age.

In addition, other factors beyond obesity, such as socioeconomic status [20, 21], culture and/or geographical location [22], mother education [23, 24], resources of the domestic environment [25], resources of the school [26] and PA levels [27–30], may influence gross MC. Thus it is crucial to match groups concerning these factors to avoid interpretation bias.

In a systematic review [31] focusing on obesity in Brazilian children, the authors stated that although there has been an increase in publications in recent years, well-controlled studies regarding the association between overweight/obesity and MC are still scarce. Furthermore, studies have failed to provide information about sample size, variance and effect estimates, statistical power, and adjustment of possible confounders in data analysis [32]. Moreover, because most of the studies include children in a wide range of age and at many stages of biological maturation [33], the results are inconclusive, limiting the validity, interpretation and extrapolation of the results for preschoolers [31]. These studies also present different methods to measure excess body fat mass, e.g., BMI [33], waist-hip circumference [15], or body fat mass measured with bioelectrical impedance [29]. Thus, to the best of our knowledge, none of these studies have measured body fat mass using dual-energy radiological absorptiometry (DEXA), the gold standard for body fat mass measurement.

In light of the above, is excess body fat mass (measured by a gold-standard method) in preschoolers associated with worse gross motor skills when matched for socioeconomic status, home and school environment, maternal education and PA levels?

The aim of this study was: 1) To compare the gross MC between overweight/obese and eutrophic preschoolers according to PA levels, sex, age, socioeconomic status, maternal education, quality of the home environment and quality of the school environment. 2) To investigate to what extent excess body fat mass explains gross motor skills in preschoolers.

## Materials and methods

This is a quantitative, exploratory, cross-sectional study approved by the Research Ethics Committee of Universidade Federal dos Vales do Jequitinhonha e Mucuri UFVJM (Protocol: 2.773.418). Data collection took place from July to December 2019. The inclusion criteria were preschoolers aging 3 to 5 years old from public schools in a Brazilian municipality. This study was part of a larger project linked to the municipal education department that included a large database of overweight and/or obese children. Parents of preschoolers from this database were contacted and invited to participate in the study. In total, seventy parents were contacted, and sixty-six agreed to participate in the study. Exclusion criteria were children who had incomplete data in the database; preterm and low birth weight infants and infants with pregnancy or parturition complications; children with signs of malnutrition or illness that interfere with growth and development or any disability; children who were at risk of being overweight or underweight and children affected with any infection during the study period (e.g., fever, influenza, or diarrhea).

The sample size was based on a pilot study with five children in each group. For sample size calculation, we used the Standard Score Locomotor of the TGMD2 and found an effect size of 0.94 using a minimum difference of 2.70 between the groups, with a standard deviation (SD) of 3.50 (G1 = 20.1, SD ±4.30 and G2 17.4, SD± 2.70). In addition, we considered a power of 90% and an alpha error of 5%, totaling 40 participants (20 participants/ group). Forty-eight preschoolers from the database were eligible to participate in this study. All participants were previously informed of all procedures of the study during a home visit by a researcher.

Weight and height measurements were taken during the study visits. The children's weight was measured to the nearest 0.1 kg using an electronic scale and height was measured to the nearest 0.5 cm using a stadiometer. The children removed their shoes and socks before stepping on the scale and were told to stand in an upright position when measuring weight and height. BMI was calculated using body weight (kg) divided by body height squared ($m^2$) and the child's birth date was also considered to the final BMI index according to the World Health Organization software [34].

Considering the high correlation between body fat mass and BMI ($r = 0.90$, $p = 0.00$), the participants were classified into groups according to BMI, following World Health Organization (WHO) recommendations. The WHO reference curves by gender and age were considered using the WHO Anthro version 3.2.2 software [34]. According to WHO guidelines, weight-for-length/height Z score should be used to assess body weight in children under five years old. Children with a BMI <1SD were considered eutrophic and assigned to Group 1, whereas children with a BMI ≥ 2SD were considered overweight or obese and assigned to Group 2.

Age, socioeconomic status, maternal education [35], PA [29], quality of the home environment and quality of the school environment [24] were collected for control. The ICC test-retest reliability of all the questionnaire used were over 0.90.

The Brazil economic classification criterion from the Brazilian Association of Research Companies was used to verify the economic status of families. This is a questionnaire that stratifies the general economic classification from A1 (high economic class) to E (very low economic class) status [36].

Home Observation for Measurement of the Environment. Inventory—Early Childhood (ECHOME). The EC-HOME [37] is an instrument for observation and measurement of the family environment. The Early Childhood (EC) version is directed at children aged three to six years. The application of the instrument is carried out through a visit that includes observation and the performance of an interview with the people responsible for the child. The instrument

consists of 55 items, which are divided into eight subscales: 1—Learning Materials; 2—Language Stimulation; 3—Physical Environment; 4—Responsivity; 5—Academic Stimulation; 6—Modeling; 7—Variety; and 8—Acceptance. Of the 55 items, 21 are based on observation of the family environment, 10 on observation and 24 on an interview, preferably with the mother. Each item in each domain is scored dichotomously (0 or 1), with a maximum score of 55 points (high scores reflect better evaluation in each domain). The sum of the raw scores of the subscales was used for analyses. The HOME Inventory has been used worldwide to evaluate the home environment in both international [38] and transcultural studies [39], presenting psychometric characteristics investigated in Brazilian preschoolers (Cronbach's Alpha = .84 for the 55 items) [40].

The quality of the school environment was assessed using the Early Childhood Environment Rating Scales (ECERS) [39], a global measure of the quality of early education for preschoolers from 2 to 5 years of age. The ECERS-R consists of 43 items divided into seven subscales (1-Space and Furnishings, 2-Personal Care Routines, 3-Language and Literacy, 4-Learning activities, 5-Interactions, 6-Program Structure, 7- Parents and staff), that are scored during live observations. A value between 1 and 7 (Likert scale) is given to each of these subscales ranging from inadequate to excellent. Then global scores are calculated based on the averages of all items. Following the standard instrument guidelines, the final score is classified as inadequate care (1.00–2.99), adequate care (3.00–4.99), or good to excellent care (5.00–6.99; Harms et al., 2013) [41]. This questionnaire is a well-known international instrument translated to the Portuguese language [42] and used by different Brazilian studies [42–44] with established psychometric properties for Brazilian preschoolers [45].

The PA level was measured using an accelerometer (Actigraph®- Model GT9X); for three valid days (excluding weekends) [46], for a minimum of 570 minutes per day [29], which is considered suitable for preschoolers [47]. Accelerometers were initialized and analyzed using 5-second epochs. In all analyses, consecutive periods of ≥ 20 minutes of zero counts were defined as non-wear time [46], with a sampling rate of 60 Hz. The accelerometer was positioned on the right side of the hip to capture accelerations and decelerations of the body and determine objective measurements of gross acceleration, PA intensity, intervals and the total time of suspension of use [46]. The classification adopted for "active" or "insufficiently active" was established according to the WHO, which considers an active child the one who has PA levels of at least 180 minutes/day, being at least 60 minutes/day of moderate to vigorous PA [48]. Then, pediatric cutoff points validated for preschoolers were used to classify the children's PA levels as insufficient (0 to 819 counts / m), mild (820 to 3907), moderate (3908 to 6111) and vigorous (above 6612)PA levels [49]. A trained researcher placed the device on the child's hip at 7 a.m. and the person responsible for the child was instructed to remove it at 7 p. m. If necessary, a new collection took place in the following week. The average time was calculated considering the simple average of the valid data.

MC was measured using the Test of Gross Motor Development second edition (TGMD-2). The reference is based on a criterion for children between three and ten years old. The TGMD-2 consists of 12 motor skills divided into two categories: 1. locomotor (run, leap, gallop, hop, jump and slide); and 2. object control (catch, strike, bounce over and underhand throw, and kick). For each skill, specific motor criteria are observed based on mature movement patterns referenced in the literature and by professionals in the field. The results obtained for each subtest are added, and the raw scores are converted into normalized scores for sex and age with a mean of 100 ± 15 [50], validated for Brazilian children [51]. For the study, the standardized scores described in Standard Score Locomotor (LP), Standard Score Object Control (OC) and Sum of the Gross Motor Quotient (GMQ- which includes the LP and OC) were used. The subtest standard scores are combined and converted to an overall

Gross Motor Quotient (GMQ), determining a child's gross motor skills compared to the tests from the standardized population. For this study, each subscale of the test (LP and OC) and the GMQ were used as dependent variables for statistical analysis. The reliability for TGMD2 showed intra-class correlation coefficients (ICC) of 0.895 for the LP, 0.925 for OC and 0.841 for GMQ. The tests were applied at the Exercise Physiology laboratory from the Universidade Federal dos Vales do Jequitinhonha e Mucuri (UFVJM), with footage of the participants for later scores by the examiner. A recent systematic review [52] indicated that, regardless of the test variation, the TMGD has moderate-to-excellent internal consistency, good-to-excellent inter-rater reliability, good-to-excellent intra-rater reliability, and moderate-to-excellent test-retest reliability.

The EC-HOME questionnaire [37] was applied at the child's home, the ECERS [41] at the school where the child studied and the other tests were applied at the Exercise Physiology laboratory from the Universidade Federal dos Vales do Jequitinhonha e Mucuri (UFVJM). All tests and measurements, including body weight, height, assessment of gross MC, as well as questionnaires, were applied by a trained examiner. The children were evaluated following a previously defined order, with an interval between data collection of a maximum of 3 weeks.

The data were analyzed using the *Statistical Package for the Social Sciences (SPSS* version 2.2). First, the Shapiro-Wilk (SW) test was performed to assess data normality, followed by Levene's test to verify the homogeneity of the variance. The descriptive statistics of continuous variables were expressed as median (minimum and maximum) or mean ± standard deviation. A Chi-square test was applied to compare the frequency of children in the eutrophic and overweight/obese groups. The t-test for independent samples (for variables with normal distribution) or the Mann-Whitney test (for variables with non-normal distribution) was used to verify differences between groups.

Spearman's or Pearson's correlation was performed to verify the relationship between body fat mass and gross motor competence, followed by the simple linear regression model between independent variables and the dependent variable LP. Variables with $p < 0.2$ was included in the multiple linear regression model analysis (stepwise). Statistical significance was set at 5%.

## Results

Table 1 presents the sample characteristics. Forty-eight children (24 eutrophic and 24 overweight) were evaluated. Among the children with excess body fat mass, 17 (70.8%) were obese and 7 (29.2%) were overweight.

The mean age in both groups of preschoolers was 4.5 years (±0.6). Most of the children belong to extract C in the economic classification. In both groups, there was a predominance of young adult mothers with completed high school education. More than half of the preschoolers attended part-time schools with a playground and other physical space for PA practices. Regarding the home environment stimulation, in the eutrophic group, 23 (95.8%) children lived in medium stimulation environment and 1 (4.2%) lived in high stimulation environment; in the overweight group, 16 (66.7%) children lived in medium stimulation environment, while 8 (33.3%) lived in high stimulation environment (Table 1).

There was no difference between groups for variables that influence children's motor behavior (age, sex, economic status, maternal education, PA, EC-HOME, quality of the school) and others that characterize them, demonstrating the pairing of the sample. As expected, the amount of body fat mass was different between the groups (p value <0.001) (Table 1).

The comparison between groups for Motor Competence are found in Table 2.

There was no difference between groups for OC subscales and GMQ. A significant difference was found in the LP between the groups; compared with the normal-weight group, the

**Table 1. Sample characteristics.**

| Variable | Eutrophic | Overweight | Test | p-value |
|---|---|---|---|---|
| | N = 24 | N = 24 | | |
| Age (years) | 4.58 ±0.58 | 4.68±0.64 | 1.151.0[b] | 0.627 |
| Sex N (%) | | | 0.19 [c] | 0.656 |
| Female | 11(45.8) | 10(41.7) | | |
| Male | 13(54.2) | 14(58.3) | | |
| Mother's age (years) | 30.96 ±6.09 | 31.88 ± 5.81 | 21.51[b] | 0.306 |
| Economic status N (%) | | | 224.50 [a] | 0.171 |
| B | 4(16.7) | 9(37.5) | | |
| C | 18(75.0) | 13(54.1) | | |
| D-E | 2(8.3) | 2(8.3) | | |
| Maternal Education N (%) | | | 279.50 [a] | 0.841 |
| Primary | 3(12.5) | 5(20.8) | | |
| Secondary | 17(70.8) | 12(50.0) | | |
| Higher | 4(16.7) | 7(29.1) | | |
| Amount of body fat mass (kg) | 4.02±1.11 | 10.89±2.96 | 1.00 [a] | <0.001 |
| Mean Intensity of PA(minutes/day) | | | | |
| Sedentary | 398.42±41.67 | 397.74±46.42 | 0.04[b] | 0.965 |
| Light PA | 190.34±37.68 | 185.08±32.94 | 0.51 [b] | 0.612 |
| Moderate PA | 37.33±10.93 | 42.02±9.51 | -1.43 [b] | 0.158 |
| Moderate to Vigorous PA | 59.17±15.45 | 61.09±14.30 | -0.34 [b] | 0.186 |
| Classification of PA N (%) | | | 0.18[c] | 0.448 |
| Active | 12 (50) | 14 (58.33) | | |
| Insufficiently active | 11 (45.8) | 10 (41.66) | | |
| EC-HOME (point) | 37.91±4.79 | 40.75±5.90 | 209.0[a] | 0.102 |
| Quality of the school environment (ECERS) (point) | 2.58±0.28 | 2.58±0.29 | 271.0[a] | 0.725 |

Data presented by mean ± standard deviation, median (min-max). PA: Physical Activity. EC-HOME: Early Childhood Home Observation for Measurement of the Environment. ECERS: Early Childhood Environment Rating Scales.

[a] Mann-Whitney U Test.

[b] T-test for independent samples.

[c] chi-square test.

overweight/obese group had lower LP scores (Table 2). The post-hoc analysis, considering an effect size of 0.70 (alpha value = 0.05), revealed a large statistical power for the LP (Power = 0.96).

**Table 2. Comparison between groups for motor competence.**

| | Eutrophic (N = 24) | Overweight (N = 24) | Difference between groups* | p-value | 95%CI |
|---|---|---|---|---|---|
| TGMD2 LP | 9.04±1.89 | 7.63±2.08 | 2.46 | 0.018 | 0.25–2.57 |
| TGMD2 OC | 8.88±2.69 | 8.33±1.73 | 0.823 | 0.412 | -0.77–1.85 |
| TGMD2 GMQ | 93.38±11.17 | 83.54±9.09 | 1.98 | 0.053 | -0.08–11.75 |

Data presented by Mean ±standard deviation or median (Minimum-Maximum).

*T-test for independent samples. Abbreviations: TGMD2 LP = Standard Score Locomotor. TGMD2 OC = Standard Score Object Control, TGMD2 GMQ = Gross Motor Quotient.

For OC there were no significant values (data presented in S1 Data). The amount of body fat mass was negatively correlated with the LS score (Pearson's correlation; r = -0.38, p-value = 0.007).

Table 3 presents data from simple linear regression. Age and body fat mass presented a p value < 0.2 and was included in the multiple linear regression model analysis. Body fat mass showed significance with LP (p = 0.007). The multiple linear regression is presented in Table 4.

Multiple linear regression analysis showed that there was an inverse relationship between body fat mass and LP. The high body fat mass explained 12% of the low values in the LP scores in preschoolers. Specifically, for each 1 kg of body fat mass increase, it is expected a 0.19 points decrease in the LP score, with a medium effect size (d = 0.16) (Table 4).

## Discussion

This study aimed to compare the MC between overweight/obese and eutrophic preschoolers matched for PA levels [29], age, socioeconomic status, maternal education [35], quality of the home environment and quality of the school environment [24]. Identifying variables that may interfere with child development has an important clinical relevance since the child's reciprocal relationships with the environment can influence child development [24, 53]. We believe that these variables did not impact the MC due to the similarities between the groups with regard to by variables described (Table 1) with non-significant p-value (p> 0.05).

The preschoolers of both groups were selected from public schools in the same municipality (Diamantina city, Minas Gerais, Brazil). Of note, the methodology of the present study followed the same methodology from our previous study, also developed with infants [12]. Excess body fat mass was the only factor associated with the worst MC in preschoolers. To the best of our knowledge, this is the first study comparing MC between eutrophic and overweight/obese preschoolers using robust methods for measuring PA levels (accelerometers) and body fat mass content (DEXA). Moreover, we collected some critical data that may interfere with the motor development in preschoolers, i.e., socioeconomic status, maternal education [35] and the quality of the home and school environment. Regarding the home environment, we evaluated the availability of resources and toys, trips and opportunities for stimulating experiences, use of free time, family routines and meetings, physical space of the home environment and the direct involvement of parents in the child's life [24, 37].

The excess of body fat mass appears as a factor that interferes with MC in LP. Being overweight/obese seems to hinder displacements since antigravity activities are more difficult [33, 54] due to the morphological restrictions to movement within high biomechanical restrictions that make it more challenging to perform tasks involving changes in the center of mass [16]. Other studies [53, 55] have also found an inverse relationship between excess body fat mass and motor skills in preschoolers [16, 29]. Studies with Brazilian children (3 to 5 years old) used the same motor tests as in the present study and also found an inverse relationship between body fat mass and LP scores [15].

Excess body fat mass is associated with worse MC [17, 18, 32, 56, 57], as overweight/obese seems to contribute to declines in motor proficiency. Despite previous study [56], investigating temporal precedence in the relationship between MC and weight status in schoolchildren aged between 5 and 10, found poor MC did not predict weight gain, the literature is conflicting. Although previous work has shown that fat mass was not related to any motor skill measure [13], according to Lima and colleagues [18], poor MC at 6 years old seems to be associated with excess body fat mass during childhood [18].

**Table 3. Regression between independent variables and standard locomotor score.** (N = 48).

| | Simple linear regression | | | | |
|---|---|---|---|---|---|
| | Locomotor subset standard score | | | | |
| Variable | ß | B | 95% CI | *p-value* | R² |
| Classification of PA | 0.140 | 0.590 | -0.66-(1.84) | 0.348 | -0.002 |
| Sex | 0.614 | -0.423 | -1.65-(0.81) | 0.494 | 0.010 |
| Age (years) | -0.211 | -0.725 | -1.72-(0.27) | 0.150 | 0.024 |
| Economic status | 0.177 | -0.348 | -0.22-(0.92) | 0.228 | 0.010 |
| EC-HOME (point) | 0.067 | 0.026 | -0.08-(0.13) | 0.649 | -0.005 |
| Quality of the school environment (ECERS) (point) | 0.055 | 0.401 | -1.76-(2.56) | 0.711 | 0.003 |
| Body fat mass(kg) | -0.382 | -0.195 | -0.334-(-0.055) | 0.007 | 0.146 |

Note: ß = standard regression coefficient; B = non-standard regression coefficient; 95% CI = 95% confidence interval; estimate of the increase or decrease of the dependent variable for each increase of one unit of the independent variable; p = statistical significance; R² = coefficient of determination. PA: physical activity. EC-HOME: Early Childhood Home Observation for Measurement of the Environment. ECERS: Early Childhood Environment Rating Scales.

Our data revealed an inverse association between excess body fat mass and MC. Here, the excess body fat mass explained 12% of the worse LP scores in preschoolers. Thus, the present study results reveal strong evidence of a negative association between excess body fat mass and MC [33] and the relative declines in children's motor proficiency can serve as a catalyst for inactivity and consequent weight gain with advancing age [10, 53]. In this sense, we emphasize that this relationship can be bidirectional from a preschool age [15].

This study has some limitations and strengths. The sample was small; however, the sample size was sufficient to achieve a medium to large effect size. The study has a cross-sectional format, which does not allow inferring a cause-and-effect relationship; then, longitudinal studies are necessary to examine the development of MC and its relationship with other health-related outcomes. However, to our knowledge, this is the first well-controlled study investigating the influence of excess body fat mass on MC in preschoolers. Among the strengths of this study, we highlight the short data collection interval (maximum of three weeks), the use of a standardized instrument for assessing MC [50], the use of a TGMD2 test validated for Brazilian children [51], the direct measurement of PA levels with accelerometer [29], and the use of DEXA, a gold-standard method to measure body fat mass [58]. Finally, we also highlight the assessment of relevant factors that interfere in the child's motor development [24, 29], i.e., socioeconomic level of the family, maternal schooling [35], quality of the school environment [24, 46] and quality of the home environment [24, 37].

## Conclusions

The excess body fat mass is associated with worse locomotor skills in preschoolers even when even when matched for other important variables that can interfere with MC. These findings

**Table 4. Multiple linear regression.**

| Standard Score Locomotor | | | | | |
|---|---|---|---|---|---|
| Variable | ß | B | 95% CI | *p-value* | R² |
| Body fat mass | -0.382 | -0,195 | -0.33– (-0.05) | 0.007 | 0.12 |

Note: ß = standard regression coefficient; B = non-standard regression coefficient; 95% CI = 95% confidence interval; estimate of the increase or decrease of the dependent variable for each increase of one unit of the independent variable; p = statistical significance; R2 = coefficient of determination

point to the importance of developing strategies to increase preschoolers' motor competence (especially locomotion skills) and reduce excess body fat mass in preschoolers.

## Practical implications

- Preschoolers with excess body fat mass have worse locomotor skills than eutrophic preschoolers even when matched for maternal education, socioeconomic status, PA, sex, age, quality of the home environment and quality of the school environment.

- The worst locomotor skills in preschoolers with excess body fat mass reveal the importance of strategies to stimulate locomotor skills, especially in the context of pediatric obesity.

## Supporting information

**S1 Data.**
(XLSX)

**S1 File. Regression between independent variables and standard score object control (N = 48).**
(DOCX)

## Acknowledgments

The authors are grateful to the municipal education secretary and directors of public schools of Diamantina (MG).

## Author Contributions

**Conceptualization:** Juliana Nogueira Pontes Nobre, Rosane Luzia De Souza Morais, Amanda Cristina Fernandes, Ângela Alves Viegas.

**Data curation:** Juliana Nogueira Pontes Nobre, Rosane Luzia De Souza Morais, Amanda Cristina Fernandes, Ângela Alves Viegas.

**Formal analysis:** Juliana Nogueira Pontes Nobre, Rosane Luzia De Souza Morais, Amanda Cristina Fernandes, Ângela Alves Viegas.

**Investigation:** Juliana Nogueira Pontes Nobre, Ângela Alves Viegas, Pedro Henrique Scheidt Figueiredo.

**Methodology:** Juliana Nogueira Pontes Nobre, Rosane Luzia De Souza Morais, Ângela Alves Viegas, Pedro Henrique Scheidt Figueiredo, Henrique Silveira Costa.

**Resources:** Pedro Henrique Scheidt Figueiredo, Ana Cristina Resende Camargos.

**Software:** Pedro Henrique Scheidt Figueiredo, Henrique Silveira Costa.

**Supervision:** Vanessa Amaral Mendonça, Ana Cristina Rodrigues Lacerda.

**Validation:** Ana Cristina Rodrigues Lacerda.

**Visualization:** Ana Cristina Resende Camargos, Ana Cristina Rodrigues Lacerda.

**Writing – original draft:** Juliana Nogueira Pontes Nobre, Rosane Luzia De Souza Morais, Vanessa Amaral Mendonça, Ana Cristina Rodrigues Lacerda.

**Writing – review & editing:** Ana Cristina Resende Camargos, Marco Fabrício Dias-Peixoto, Vanessa Amaral Mendonça, Ana Cristina Rodrigues Lacerda.

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
