## [Decision Letter · Decision Letter 0]

13 Oct 2021

PONE-D-21-16031

Is body fat mass associated with worse gross motor skills in preschoolers? An exploratory study

PLOS ONE

Dear Dr. Nobrr,

Thank you for submitting your manuscript to PLOS ONE. After careful consideration, we feel that it has merit but does not fully meet PLOS ONE’s publication criteria as it currently stands. Therefore, we invite you to submit a revised version of the manuscript that addresses the points raised during the review process.

The manuscript has been evaluated by two reviewers, and their comments are available below.

The reviewers have raised a number of concerns regarding the manuscript’s clarity and organization. They specifically request a more thorough and comprehensive view of the literature in the Introduction/Discussion, as well as more methodological reporting details. They additionally request significant language editing, which may help to address the above points.

Could you please carefully revise the manuscript to address all comments raised?

We look forward to receiving your revised manuscript.

Kind regards,

Avanti Dey, PhD

Staff Editor

PLOS ONE

2. Please state whether you validated the questionnaire prior to testing on study participants. Please provide details regarding the validation group within the methods section.

3. Thank you for submitting the above manuscript to PLOS ONE. During our internal evaluation of the manuscript, we found significant text overlap between your submission and the following previously published works, some of which you are an author.

- https://www.researchsquare.com/article/rs-612416/v1

- https://www.researchgate.net/publication/283530031_Test_of_gross_motor_development-2

- https://www.frontiersin.org/articles/10.3389/fphys.2019.01273/full   (last paragraph of discussion) 

Please revise the manuscript to rephrase the duplicated text, cite your sources, and provide details as to how the current manuscript advances on previous work. Please note that further consideration is dependent on the submission of a manuscript that addresses these concerns about the overlap in text with published work.

Additional Editor Comments (if provided):

Reviewers' comments:

Reviewer's Responses to Questions

**Comments to the Author**

1. Is the manuscript technically sound, and do the data support the conclusions?

Reviewer #1: No

Reviewer #2: No

2. Has the statistical analysis been performed appropriately and rigorously? 

Reviewer #1: No

Reviewer #2: No

3. Have the authors made all data underlying the findings in their manuscript fully available?

Reviewer #1: Yes

Reviewer #2: Yes

4. Is the manuscript presented in an intelligible fashion and written in standard English?

Reviewer #1: No

Reviewer #2: Yes

5. Review Comments to the Author

Reviewer #1: Review Ms PONE-D-21-16031

Is body fat mass associated with worse gross motor skills in preschoolers? An exploratory study.

The authors compared motor competence in a relatively small sample of overweight/obese and eutrophic children (5 yrs of age), thereby taking a series of secondary variables (PA, sex, gender, age, SES, etc…) into account. While the underlying rationale of the study is to be applauded, and the authors collect an impressive set of variables with state-of-the-art assessment instruments (great effort, much appreciated!!!), the writing of the paper itself is problematic and unacceptable for publication in this form. I had the impression that this paper was written under severe time pressure, resulting in a kind of ‘quick & dirty job’.

I hope that my comments in attachment are helpful for the authors to rework this material that does have publication potential, but not in its current form.

Reviewer #2: The manuscript entitled “Is body fat mass associated with worse gross motor skills in preschoolers? An exploratory study” compared the motor competence of overweight/obese preschoolers with eutrophic peers. Overall, the manuscript is well written and it approaches an important topic regarding child health. However, there are some issues in this work which should be better addressed by the authors. Both general and specific comments are provided as per below.

General comments

Literature provides extensive evidence about the relationship between motor competence and weight status across childhood, yet the background provided by the authors is little informative. As such, it is not clear what is already known about the relationship between body fat and motor competence in preschool children. In addition, the authors argued that “the strength of the present study is the consideration of multiple factors that influence child development” (page 4). However, it is not clear the rationale for considering these multiple factors (i.e. PA, sex, age, socioeconomic status, maternal education, quality of the home environment and quality of the school environment) as potential confounders. In addition, it does not seem appropriate to consider “factors that influence child development” in a cross-sectional study.

Also, this cross-sectional study has a small sample size, especially when considering the multiple factors/variables inserted in the regression analysis.

Specific comments

Introduction

- Overall, I would suggest to provide a focused background considering the relationship between weight status and motor competence across childhood. This seems the main focus of this study and as such it should be better presented in the Introduction.

- The state of knowledge about the relationship between weight status and motor competence is missing. Please, provide a more detailed background and explain the ‘why’ we should consider “PA, sex, age, socioeconomic status, maternal education, quality of the home environment and quality of the school environment” as potential confounders.

Methods

“For sample calculation, a power of 90%, and an alpha error of 5% were considered, with 20 participants thus being required for each group, totaling 40 subjects.” (page 5, 2nd paragraph).

And the effect size? Did you calculate the sample size considering the number of total predictors and the effect size?

Results

In the Table 3, there two regression models (i.e. pre- post adjustment). Please, explain this analysis in the Methods section.

Discussion

I think the Discussion should be reassessed by the authors considering the previous comments.

6. PLOS authors have the option to publish the peer review history of their article (what does this mean?). If published, this will include your full peer review and any attached files.

Reviewer #1: No

Reviewer #2: No

---

## [Author Response · Author response to Decision Letter 0]

18 Nov 2021

Ref: PONE-D-21-16031

Title: Is body fat mass associated with worse gross motor skills in preschoolers? An exploratory study

PLOS ONE

Dear Avanti Dey, PhD

Staff Editor

PLOS ONE

We appreciate the reviewer’s comments. They were important to improve the quality of our manuscript. See below the point-to-point answers.

PONE-D-21-16031

Answer: We have made the requested adjustments.

2. Please state whether you validated the questionnaire prior to testing on study participants. Please provide details regarding the validation group within the methods 

Answer: The ICC test-retest reliability of all the questionnaires used was over 0.90 (see line 136). We enter the psychometric properties of the instruments used in the Methods session 

3. Thank you for submitting the above manuscript to PLOS ONE. During our internal evaluation of the manuscript, we found significant text overlap between your submission and the following previously published works, some of which you are an author.

- https://www.researchsquare.com/article/rs-612416/v1

- https://www.researchgate.net/publication/283530031_Test_of_gross_motor_development-2

- https://www.frontiersin.org/articles/10.3389/fphys.2019.01273/full (last paragraph of discussion) 

Please revise the manuscript to rephrase the duplicated text, cite your sources, and provide details as to how the current manuscript advances on previous work. Please note that further consideration is dependent on the submission of a manuscript that addresses these concerns about the overlap in text with published work.

Answer: We have made the requested adjustments. The description of the TGMD2 test has been duly cited (see URICH, 2000). Of note, Plos One team encourage to researchers to submit their manuscript to a preprint server directly (https://plos.org/open-science/preprints/preprint-faqs/). Thus, we have made a preprint submission. In addition, despite our team having another study (https://www.researchsquare.com/article/rs-612416/v1) in line with the present study, they are very different, including the question. With this regard, we have used an anti-plagiarism software Copyspider (Copyspider), obtaining a value of 2.43% (up to 3% is acceptable). Finally, we have carefully reviewed the manuscript before resubmission.

Reviewer #1: 

Review Ms PONE-D-21-16031

Is body fat mass associated with worse gross motor skills in preschoolers? An exploratory study.

We thank the reviewer for the comments and suggestions; they helped improve the quality of our manuscript further.

Language: This manuscript needs a thorough revision by a native English speaker or by a professional familiar with the writing of scientific reports.

Answer: All the authors, some of them living abroad, have carefully reviewed the English language. 

Abstract: • Use the same tense during the abstract • ‘quantitative’: how to be interpreted here? Should be quantitatively? • Sentence with ‘but similar skills’… check grammar • Globale language revision is mandatory

Answer: We have corrected the tense and grammar throughout the text. 

Introduction: • The introduction is too short and superficial to allow the development of a solid rationale for this study. Specifically with respect to the weight/motor skill relationship, the authors can start from the abundance of literature in children from 5 years and older, describe what we know already in more detail (point to the urgency of the problem), and then move on tot he part in which the paucity of similar studies in younger children is described. This approach will also help to formulate specific hypotheses at the end of the introduction. You may want to consults the publications of Eva D’Hondt / Mireille Augustijnen between 2008-2010 for that purpose.

Answer: We agree with the reviewer’s comment. Because of the lack of studies with preschoolers, we have adjusted the introduction session to bring theoretical references, providing the basis for understanding how far the literature has advanced the relationship between motor competence and body fat in children. We emphasize that we were careful to pair preschool children in both groups considering the variables influencing child development and could cause interpretation bias in the primary variables (see the previous study of our team including infants up to 2 years of age, Camargos et al., 2016). (See lines 44-50; 64-68; 69-78; 82-83.)

• I applaud the inclusion of the impressive set of secondary variables, but the rationale on why to include one variabele (and not other ones) is lacking. Suggestion to shortly describe for each of the variables whether they might have a positive or a negative effect on motor competence. Again, this will help building a solid set of hypotheses.

Answer: We have improved the introduction session according to the reviewer's suggestion. (See lines 44-50; 64-68; 69-78; 82-83).

• Describing the strength of the study does not belong at the end of the introduction

Answer: We have removed this sentence from the end of the introduction session. 

Methods: • More information on the recruitment procedure is needed. How many parents were contacted for each group and how many agreed participation of their child?

Answer: We have included additional information on the recruitment procedure (see lines 105-106).

• I assume you mean that you excluded participants that were subject to an infectious process at the time of the study (not …had been subject..)? ]

Answer: Yes! We have excluded participants that were subject to any infectious process during the study. We included this information in the text (see line 110). “overweight or underweight and children affected with any infection during the study period …”

• I agree that the sample size is large enough for the straightforward comparison between groups, but did the authors also reck on with the extended set of secondary variables in their power analysis?

Answer: Although we have adjusted the final model for all variables that could interfere with the result, only "body fat mass" remained as an independent variable associated with “Standard Score Locomotor” in the model (Table 4). With this regard, we have performed the post hoc power analysis and obtained a medium effect size (d = 0.16) and a power of 0.77 (see Lines 282-285).

Print Screen:

• How is age-specific BMI age specific as you just calculate weight/m²? Please be more specific. 

Answer: To calculate the BMI, we have used a software from the World Health Organization considering the age and the child's weight and height. We have inserted this additional information and reference (see ref 32: WHO, 2011).

Lines 125-126: Age-specific BMI was calculated using a specific World Health Organization software. Of note, the calculation considered the child's date of birth and data on body weight (kg) divided by body height squared (m2).

• Please provide reliability/validity information on each test instrument.

Answer: We have provided reliability/validity information on each test instrument in the text as follows.

HOME: (See lines 153-156): The HOME Inventory has been used worldwide to evaluate the home environment in both international [36] and transcultural studies [37], presenting psychometric characteristics investigated in Brazilian preschoolers sample, analysis of internal consistency satisfactory for the total scale (Cronbach's Alpha = .84 for the 55 items) [38].

ECERS: (See lines 166-168): This questionnaire is a well-known international instrument translated to Portuguese [40] and used by different Brazilian studies with preschoolers [40; 41; 42] with established psychometric properties for Brazilian preschoolers [43].

TGMD2: (See lines 203-206): Recent study systematic review [51] indicated that, regardless of the variant of the test, the TMGD has moderate-to-excellent internal consistency, good-to-excellent inter-rater reliability, good-to-excellent intra-rater reliability, and moderate-to-excellent test-retest reliability.

• ACtigraph: more details are needed. Were the 3 days randomly chosen? Weekdays and/or weekend days? Please first describe the instrument, then how it is positioned, and then information on the analysis. Not clear how you can capture heart rate with a hip-mounted device. How is ‘the child’s mean time’ calculated? 

Answer: We have included additional information about ACtigraph measures.

(See line 170): three valid days (excluding weekends)

(See lines 181-185): A trained researcher placed the device on the child's hip at 7 a.m. and the person responsible for the child was instructed to remove it at 7 p.m. If necessary, a new collection took place in the following week. The average time was calculated considering the simple average of the valid data.

.• TGMD-2: 12 motor skills divided into two categories (not subtests) 

Answer: Yes! We have modified accordingly (see line 188): divided into two categories

• Data are presented as mean or median, as appropriate. I assume you mean ‘depending on the outcome of the SW and L test. See comment under results.

Answer: We have performed new statistical analyzes and restructured the data presentation (see table 1). Noteworthy, both KH and SW have the same distribution (non-normal) for the dependent variable. We have adjusted the data presentation according to this distribution.

• The statistics section lacks structure, and this reflects in the Tables. It is very hard to figure out what test was use for what variable from the tables. I am not convinced that it is necessary to strictly adhere to the results of the SW and L test to decide on the statistical approach for each variable independently given the –in my opinion-sufficiently large number of participants per group.

Answer: We have performed new statistical analyses and adjusted the text (see lines 222-224). “Spearman's or Pearson’s correlation was performed to verify the relationship between body fat mass and gross motor competence variables, followed by the simple linear regression model between independent variables and the dependent variable LP. Variables with p<0.2 became part of the multiple linear regression model (stepwise).Statistical significance was set at 5%.”

Results: 

• Table 1: please provide the units for each variable (eg under ‘sex’ it is N, with the proportion in brackets; is amount of body mass in kg or a %; etc.)

Answer: We have provided the units for each variable in table 1 (highlighted in the text). 

• Subscript ‘a’ is floating in the table, where does it belong to? • I would say that children this age ‘attend’ school rather than ‘study’ 

Answer: We have modified accordingly (highlighted in the text). 

• Of course it makes sense that body mass differed between groups, given that children were selected on being overweight. 

Answer: We have performed new statistical analyses (n=48, 24 in each group), clarifying a strength of our study since we collected data of variables that could lead to an interpretation bias. 

Thus, the strict internal control of our study enabled the comparison between the locomotor skills of preschoolers in similar conditions of child development, except for excess body weight. Now, we hope it is clear throughout the text.

• Table 2: mixing between parametric and non-parametric stats in subscores of the same test battery is confusing 

Answer: Yes! We agree with this reviewer’s comment. We have made the adjustments in table 2.

• Please provide also the results of the OC analyses. Not finding significant differences is as important as reporting the significant ones!

Answer: We have inserted the results of the OC analyses as a supplementary file. We also have included a paragraph about OC results in the results section.

Line 261: For OC there were no significant values (data presented in supplementary material).

Discussion: • The last paragraph of the results is confusing: first you mention the MLR adjusted for confounding factors. You end the paragraph with …’Finally, control variables were inserted in the model….’ Isn’t that twice the same? 

Answer: We agree with this reviewer’s comment that we have already mentioned that the control variables were inserted (adjusted) in the regression model. We have excluded this sentence from the results session.

• The participants were selected from public schools in the same municipality, which might explain the absence of differences in the secondary variables? Discuss where this group is to be situated compared to rest of the Brazilian children?

Answer: We have included in the discussion session the requested information as follows:

(see lines 296-299): The preschoolers of both groups were selected from public schools in the same municipality (Diamantina city, Minas Gerais, Brazil). Of note, the methodology of the present study followed the same methodology from our previous study, also developed with infants [11]. Excess body fat mass was the only factor associated with the worst MC in preschoolers.

• The results could be better framed within the knowledge on children that are slightly older. For example, several studies did find differences in fine motor skills in young obese children (I think D’Hondt et al in Neuroscience Letters or APAQ somewhere in 2008 or 2009). Might be worthwile discussing in a broader context. 

Answer: We have included additional information from recent studies investigating motor skills in young obese children (see lines 317-326): 

“Excess body fat mass is associated with worse MC [15;16;54;55;30], as overweight/obese seems to contribute to declines in motor proficiency. Despite previous study [54], investigating temporal precedence in the relationship between MC and weight status in schoolchildren aged between 5 and 10, found poor MC did not predict weight gain, the literature is conflicting. Thus, according to Lima and colleagues [16], poor MC at 6 years old seems to be associated with excess body fat mass during childhood [16].

Our data revealed an inverse association between excess body fat mass and MC. Here, the excess body fat mass explained 12% of the worse LP scores in preschoolers. Thus, the present study results reveal strong evidence of a negative association between excess body fat mass and MC [30]

• On p 14, I cannot understand why the concept of ‘body image’ suddenly emerges in the discussion. Difficult sentence to grab, please clarify.

Answer: Yes! We have excluded this information.

 • You may want to consult the recent meta-analysis by Barnett et al in Sports Medicine (2021), overviewing among others the longitudinal relationship / causality between weight status and MC (and the reverse). 

Answer: Thanks to the reviewer! We have included additional information to improve the discussion session.

In addition, lines 324-326 “Thus, the results of the present study add to the current literature, as showed a strong evidence of a negative association between excess body fat mass and MC [30]”

• Near the end of the discussion and practical implications: suggestion to stick to the findings of this study. This was a cross-sectional study, so the conclusion that excess body fat mass influences competence in locomotor skills should be toned down a bit.

Answer: We have modified the conclusion session as suggested (see lines 342-346).

The excess body fat mass is associated with worse locomotor skills in preschoolers even when even when matched for other important variables that can interfere with MC. These findings point to the importance of developing strategies to increase preschoolers' motor competence (especially locomotion skills) and reduce excess body fat mass in preschoolers.

.• The practical implications do not relate to the findings of this study at all, please revise.

Answer: We have revised the practical implications according to the findings of our study.

Reviewer #2: 

The manuscript entitled “Is body fat mass associated with worse gross motor skills in preschoolers? An exploratory study” compared the motor competence of overweight/obese preschoolers with eutrophic peers. Overall, the manuscript is well written and it approaches an important topic regarding child health. However, there are some issues in this work which should be better addressed by the authors. Both general and specific comments are provided as per below.

We thank the reviewer for the comments and suggestions; they helped improve the quality of our manuscript further.

General comments

Literature provides extensive evidence about the relationship between motor competence and weight status across childhood, yet the background provided by the authors is little informative. As such, it is not clear what is already known about the relationship between body fat and motor competence in preschool children. In addition, the authors argued that “the strength of the present study is the consideration of multiple factors that influence child development” (page 4). However, it is not clear the rationale for considering these multiple factors (i.e. PA, sex, age, socioeconomic status, maternal education, quality of the home environment and quality of the school environment) as potential confounders. In addition, it does not seem appropriate to consider “factors that influence child development” in a cross-sectional study.

Also, this cross-sectional study has a small sample size, especially when considering the multiple factors/variables inserted in the regression analysis.

Answer: Suggestion accepted! Because of the lack of studies with preschoolers, we adjusted the entire introduction session to bring theoretical references, providing the basis for understanding how far the literature has advanced the relationship between motor competence and body fat in children. We emphasize that we were careful to pair preschool children in both groups considering the variables influencing child development and could lead to interpretation bias in the primary variables [see the previous study of our team including infants up to 2 years of age (Camargos et al., 2016)]. Of note, we calculated our sample size based on a pilot study with five preschoolers in each group, totaling ten subjects. For sample size calculation, we used the Standard Score Locomotor of the TGMD2 and found an effect size of 0.94 using a minimum difference of 2.70 between the groups, with a standard deviation (SD) of 3.50. (G1=20.1, SD ±4.30 and G2 17.4, SD± 2.70). In addition, we considered a power of 90%, and an alpha error of 5%, being 20 participants required for each group, totaling 40 subjects. For the present study, 48 preschoolers from the database were eligible to participate in the study. Finally, only variables with p<0.2 were included in the multiple linear regression model analysis (stepwise).

Specific comments

Introduction

- Overall, I would suggest to provide a focused background considering the relationship between weight status and motor competence across childhood. This seems the main focus of this study and as such it should be better presented in the Introduction.

Answer: Suggestion accepted! We have provided a specific background considering the relationship between weight status and motor competence across childhood (see lines 44-50, 64-78).

- The state of knowledge about the relationship between weight status and motor competence is missing. Please, provide a more detailed background and explain the ‘why’ we should consider “PA, sex, age, socioeconomic status, maternal education, quality of the home environment and quality of the school environment” as potential confounders.

Answer: Suggestion accepted! Introduction session (lines 64-78). 

Methods

“For sample calculation, a power of 90%, and an alpha error of 5% were considered, with 20 participants thus being required for each group, totaling 40 subjects.” (page 5, 2nd paragraph).

And the effect size? Did you calculate the sample size considering the number of total predictors and the effect size?

Answer: Yes! We calculated our sample size considering the number of total predictors and calculated the effect size according to a pilot study with five preschoolers in each group, totaling ten subjects. For sample size calculation, we used the Standard Score Locomotor of the TGMD2 and found an effect size of 0.94 using a minimum difference of 2.70 between the groups, with a standard deviation (SD) of 3.50. (G1=20.1, SD ±4.30 and G2 17.4, SD± 2.70). In addition, we considered a power of 90%, and an alpha error of 5%, being 20 participants required for each group, totaling 40 subjects. For the present study, 48 preschoolers from the database were eligible to participate in the study.

See below the Print Screen of our sample size calculation. 

Results

In the Table 3, there two regression models (i.e. pre- post adjustment). Please, explain this analysis in the Methods section.

Answer: To clarify, we have inserted table 4 (multiple linear regression) including the statistical model. Moreover, we have inserted in additional information in Materials and methods 

…simple linear regression model between independent variables and the dependent variable LP. Variables with p<0.2 became part of the multiple linear regression model (stepwise).

Discussion

I think the Discussion should be reassessed by the authors considering the previous comments.

Answer: We have revised the discussion session considering the previous comments.

---

## [Decision Letter · Decision Letter 1]

17 Jan 2022

PONE-D-21-16031R1Is body fat mass associated with worse gross motor skills in preschoolers? An exploratory studyPLOS ONE

Dear Juliana Nobre,

Thank you for submitting your manuscript to PLOS ONE. After careful consideration, we feel that it has merit but does not fully meet PLOS ONE’s publication criteria as it currently stands. Therefore, we invite you to submit a revised version of the manuscript that addresses the points raised during the review process. Overall, the manuscript was amended addressing all issues raised by reviewers, including my comments as reviewer in the first round of revision. However, I would like to suggest some papers to be considered in the manuscript, especially in the Introduction and Discussion. These articles are: Hall et al. 2018 (doi:10.3390/jfmk3040057); Wood et al. 2022 (doi: 10.1089/chi.2021.0026); Henrique et al. 2019 (doi.org/10.1002/ajhb.23364)

We look forward to receiving your revised manuscript.

Kind regards,

Daniel V. Chagas

Academic Editor

PLOS ONE

Journal Requirements:

Reviewers' comments:

Reviewer's Responses to Questions

**Comments to the Author**

1. If the authors have adequately addressed your comments raised in a previous round of review and you feel that this manuscript is now acceptable for publication, you may indicate that here to bypass the “Comments to the Author” section, enter your conflict of interest statement in the “Confidential to Editor” section, and submit your "Accept" recommendation.

Reviewer #1: All comments have been addressed

2. Is the manuscript technically sound, and do the data support the conclusions?

Reviewer #1: Yes

3. Has the statistical analysis been performed appropriately and rigorously? 

Reviewer #1: Yes

4. Have the authors made all data underlying the findings in their manuscript fully available?

Reviewer #1: Yes

5. Is the manuscript presented in an intelligible fashion and written in standard English?

Reviewer #1: Yes

6. Review Comments to the Author

Reviewer #1: (No Response)

7. PLOS authors have the option to publish the peer review history of their article (what does this mean?). If published, this will include your full peer review and any attached files.

Reviewer #1: No

---

## [Author Response · Author response to Decision Letter 1]

31 Jan 2022

Ref: PONE-D-21-16031

Title: Is body fat mass associated with worse gross motor skills in preschoolers? An exploratory study

PLOS ONE

Dear Daniel V. Chagas

Academic Editor 

PLOS ONE

We appreciate the reviewer’s comments. They were important to improve the quality of our manuscript. See below the point-to-point answers.

PONE-D-21-16031

1) Overall, the manuscript was amended addressing all issues raised by reviewers, including my comments as reviewer in the first round of revision. However, I would like to suggest some papers to be considered in the manuscript, especially in the Introduction and Discussion. These articles are: Hall et al. 2018 (doi:10.3390/jfmk3040057); Wood et al. 2022 (doi: 10.1089/chi.2021.0026); Henrique et al. 2019 (doi.org/10.1002/ajhb.23364)

A) Hall et al. 2018 (doi:10.3390/jfmk3040057

Answer: We appreciate the suggestion. We inserted the reference in the introduction (line 60) “healthy behaviors that may influence physical activity (PA) practices throughout life [10, 11]”.

B) Wood et al. 2022 (doi: 10.1089/chi.2021.0026)

Answer: We appreciate the suggestion. We inserted the reference in the introduction (see lines 67-68: “Although previous work has shown that fat mass was not related to any motor skill measure [13]” and lines 322-323) “Although previous work has shown that fat mass was not related to any motor skill measure [13]”.

C) Henrique et al. 2019 (doi.org/10.1002/ajhb.23364)

We appreciate the suggestion. We inserted the reference in the introduction (see line 69: including a work with Brazilian preschoolers from the Northeast region of Brazil [15]), and in the discussion (see lines 329-330: In this sense, we emphasize that this relationship can be bidirectional from a preschool age [15]). Of note, the suggested article was already listed in the references (see reference 15).

Thank you for contacting us and we are at your disposal for any questions you may have.

Sincerely,

Juliana Nobre, PhD

Universidade Federal dos Vales do Jequitinhonha e Mucuri, Brazil

---

## [Editor Report · Decision Letter 2]

7 Feb 2022

Is body fat mass associated with worse gross motor skills in preschoolers? An exploratory study

PONE-D-21-16031R2

Dear Juliana Nobre,

We’re pleased to inform you that your manuscript has been judged scientifically suitable for publication and will be formally accepted for publication once it meets all outstanding technical requirements.

Kind regards,

Daniel V. Chagas

Guest Editor

PLOS ONE
---

## [Editor Report · Acceptance letter]

15 Feb 2022

PONE-D-21-16031R2 

Is body fat mass associated with worse gross motor skills in preschoolers? An exploratory study 

Dear Dr. Nobre:

I'm pleased to inform you that your manuscript has been deemed suitable for publication in PLOS ONE. Congratulations! Your manuscript is now with our production department. 

Kind regards, 

on behalf of

Dr. Daniel V. Chagas 

Guest Editor

PLOS ONE